# Effect of Grain Size on Mechanical and Creep Rupture Properties of 253 MA Austenitic Stainless Steel

Mochammad Syaiful Anwar [1,2,*], Robert R. Widjaya [3], Leonardo Bayu Adi Prasetya [4], Abdul Aziz Arfi [5], Efendi Mabruri [2] and Eddy S. Siradj [1]

1. Department of Metallurgical and Materials Engineering, Faculty of Engineering, Universitas Indonesia, Kampus UI, Depok 16424, Indonesia; eddy.sumarno@ui.ac.id
2. Research Center for Metallurgy, National Research and Innovation Agency, Kawasan PUSPIPTEK Building 470, Tangerang Selatan 15314, Indonesia; efendi_lipi@yahoo.com
3. Research Center for Chemistry, National Research and Innovation Agency, Kawasan PUSPIPTEK Building 452, Tangerang Selatan, Banten 15314, Indonesia; robe007@brin.go.id
4. PT. Pembangkit Jawa Bali, Unit Maintenance Repair dan Overhaul, Muara Karang, Jakarta Utara 14450, Indonesia; leonardo.bayu@ptpjb.com
5. PT. Robutech, Technical Testing and Inspection Services, Surabaya 60119, Indonesia; aziz@robutech.com
* Correspondence: moch031@brin.go.id

**Abstract:** The effect of grain size on the mechanical properties and creep rupture of 253 microalloyed (MA) austenitic stainless steel (ASS) was investigated. The cold rolling process with a 53% reduction in thickness was applied to the steel followed by annealing at 1100 °C over 0, 900, 1800, and 3600 s to obtain grain sizes of 32.4, 34.88, 40.35, and 43.77 μm, respectively. Uniaxial tensile and micro-Vickers hardness tests were carried out to study the effect of grain size on mechanical properties at room temperature. The creep rupture test was performed at 700 °C under a load of 150 MPa. The results showed that there was a correlation between grain size, mechanical properties, and creep rupture time. The fine initial grain size showed relatively good mechanical properties with a short creep rupture time, while the coarse initial grain size produced low mechanical properties with a long creep rupture time. The initial grain size of 40.35 μm was the optimum grain size for a high value of creep rupture time due to the low hardness and elongation values at room temperature and low creep ductility value. The intergranular fracture was found on the initial grain size below 40.35 μm, and a mixed mode of intergranular and transgranular fracture was found on the initial grain size above 40.35 μm after the creep rupture test.

**Keywords:** grain size; mechanical properties; creep rupture; 253 MA; austenitic stainless steel





## 1. Introduction

Grain boundary strengthening in metal is one of the efforts made to improve mechanical properties. Work hardening, followed by annealing, is one method to change the grain size of the metal. During annealing, the recovery, recrystallization, and grain growth processes can produce different grain sizes, which are affected by the annealing temperature and time. A fine grain size can increase strength, decrease ductility, and increase metal hardness, and vice versa [1]. In the recrystallization process, both statically and dynamically, the degree of plastic deformation and strain induce martensite to affect the level of the fineness of the grain size and mechanical properties [2]. Several cold deformation techniques have been described in previous studies, followed by annealing to obtain fine grain sizes [3–6]. The presence of carbide precipitates, intermetallic phases, and micro-alloying present either in the austenite matrix or at the grain boundaries and annealing twins that appear during annealing are also believed to affect the mechanical properties [7–10].

Metals exposed to high temperatures impact the grain size to be large or coarse, which causes the strength of the metal to decrease. However, the large grain size gives

it an advantage over the creep properties of metals. Creep is a deformation indicated by increasing strain as a function of time that can occur when the metal is exposed to high temperatures and constant loads [1]. In the primary stage, the strain increases drastically for a short time and then approaches constantly in the secondary stage. In the secondary stage, the creep mechanism can occur in grain boundary sliding (GBS), diffusion creep (Nabarro–Hearring diffusion creep and Coble creep), and dislocation creep (Harper–Dorn dislocation creep) [11,12]. The mechanism is theoretically equally influenced by grain size [12–14]. Ruano et al. [12] stated that critical grain size is influenced by dislocation density and stacking fault energy that distinguishes the occurrence of diffusion creep and dislocation creep mechanisms.

Meanwhile, Galindo-Nava et al. [14] stated that GBS and dislocation creep occur in different grain size ranges. Coarse grain size generally results in a lower minimum creep rate than refined grains so that the fracture time increases [6,13]. However, Liu et al. [15] reported that large grain size does not always affect the creep rupture time. Then, the strain increases rapidly in the tertiary creep stage and ends with fracture. Surface fracture due to the fact of creep can occur in a brittle, ductile, or ductile–brittle mixture [11,16,17]. Wang et al. [11] found that when creep was applied to IN617 at a temperature less than 950 °C and stress of $0.12\sigma TS$ and $0.2\sigma YS$, it resulted in a mixed ductile–brittle fracture. Morris et al. [16] stated that at high creep temperatures, 316 ASS experienced high creep ductility and short fracture times, and the creep ductility was low when the steel was exposed to low creep temperatures due to the presence of an intergranular carbide and intermetallic phase. Wei et al. [17] stated that the A286 alloy could fracture from ductile to brittle when the temperature decreases and the creep stress increases.

253 microalloyed (MA) (UNS S30815) is a high-temperature austenitic stainless steel (ASS) widely applied in the power, petrochemical, and metallurgical industries. This steel is a variant of EN 1.4828 with a high nitrogen content and is micro-alloyed with the rare earth metal (REM) cerium. This steel is suitable for use at high temperatures from 850 to 1100 °C, has carburizing resistance, and has sufficient creep strength. This steel is not suitable for use between 600 and 850 °C, because it can cause a decrease in the impact toughness value at room temperature [18,19]. Several studies have focused on the properties of resistance to fatigue [20], oxidation, high-temperature corrosion, and flow softening of this steel [21]. However, to the best of the authors' knowledge, there has been no study on the effect of grain size on the mechanical and creep properties of 253 MA ASS, except Maode et al. [22] who reported the effect of cerium on the creep properties of 253 MA ASS.

As previously mentioned, grain growth kinetics, grain size, and annealing twins affected the hardness values of 253 MA ASS after low reduction by multi-pass cold rolling and continued with annealing [23,24]. In this study, the 253 MA ASS pipe was reduced by approximately 53% using multi-pass cold rolling and then annealed with variations in time to obtain different austenite grain sizes. The purpose of this research was to determine the effect of grain size on the mechanical and creep properties, which are applicable for early failure prevention of components exposed to high temperatures through mechanical degradation and creep rupture time prediction by simply measuring grain size.

## 2. Materials and Methods

The material used in this research was a 253 MA ASS pipe with an outside diameter of 60.33 mm and a length of 200 mm. The chemical composition is shown in Table 1.

**Table 1.** Chemical composition of 253 MA ASS (%wt.).

| C | Si | Mn | P | S | Cr | Ni | N | Ce | La | Fe |
|---|----|----|---|---|----|----|---|----|----|----|
| 0.079 | 1.422 | 0.51 | 0.03 | <0.005 | 22.06 | 10.86 | 0.384 | 0.03 | 0.014 | Bal. |

The steel was cut using an Electrical Discharge Machining (EDM)wire cutting machine (Taizhou Jiangzhou CNC Machine Tool company, Jiangsu, China) to form a cold-rolled

sample with a length of 165 mm, a width of 25.7 mm, and a thickness of 3.9 mm. Cold rolling was carried out to reduce the thickness of the sample to approximately 53% so that the final dimensions achieved were a length of 267 mm, a width of 26 mm, and a thickness of 2.3 mm. The cold-rolled sample was then formed into a tensile test sample according to American Society for Testing and Materials (ASTM) E8 and a creep rupture test specimen using an EDM wire cutting machine. The dimensions of the creep test specimen can be seen in Figure 1.

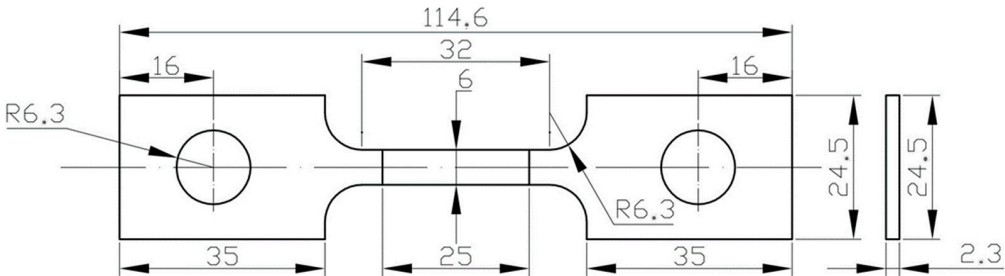

**Figure 1.** Creep rupture test sample (unit: mm).

The samples were annealed at 1100 °C for 900, 1800, and 3600 s with a $H_2$ gas atmosphere and then cooled in the cold zone of the tubular furnace (Nabertherm GmbH RSH 50, Lilienthal, Germany) to obtain grain sizes of 32.4, 34.88, 40.35, and 43.77 μm. The technical specifications of the cold rolling, tubular furnace, and annealing processes are explained in References [23,24].

The uniaxial tensile test was carried out at room temperature using a hydraulic universal testing machine (UTM) by Tinius Olsen, Horsham, PA, USA. The creep rupture test was carried out at a temperature of 700 °C and a load of 150 MPa using ZwickRoell test equipment (ZwickRoell GmbH & Co. KG, Ulm, Germany). The microstructures were observed on the samples before and after creep testing using an AmScope MIUI 1803 microscope (AmScope.com, Irvine, CA, USA). The samples were ground using SiC paper with grits of 200, 400, 600, 1000, and 2000 and then polished using diamond paste with sizes of 5, 3.5, and 1 nm. An electrolytic etch was applied to the samples in a 25% oxalic acid solution under a potential of 9 V. The grain size was measured using a measurement method of line intercept according to ASTM E112. ImageJ software (Version 1.53r, Berkeley Software Distribution (BSD) licenses, Oakland, California, USA) was employed to measure grain size. The micro-Vickers Mitutoyo hardness test (Mitutoyo America Corporation, Aurora, Illinois, USA) under a load of 0.3 N was used to measure the hardness of the sample 253 MA ASS, which consisted of different grain sizes. Scanning Electron Microscope (SEM) observations were carried out to observe the fracture surface of the tensile and creep test samples, and Energy Dispersive X-ray Analysis (EDAX) analysis was also conducted to observe the precipitates using the JEOL Type JSM 6390 A test equipment (JEOL, Ltd, Tokyo, Japan).

## 3. Results and Discussion

### 3.1. Grain Growth of 253 MA ASS after 53% Reduction

Figure 2 shows the grain growth of 253 MA ASS during annealing at 1100 °C with variations in annealing time. In this figure, the grain size increased slightly with holding time. In addition, this picture shows a comparison of grain sizes obtained from the experiments and predictions. The power law equation [25] predicts the grain size shown in Equation (1).

$$d^n - d_0{}^n = k \cdot t \qquad (1)$$

where $d$ is the average grain diameter, $d_0$ represents the initial grain size, $n$ is the growth exponent, $t$ is the soaking time, and $k$ is the growth coefficient.

The grain sizes resulting from the calculations using Equation (1) showed values close to the experimental results with an error of approximately 1–4%. Constants k and n were

obtained from calculations using solve-excel. A value of $n = 1$ indicates that normal grain growth occurred [26].

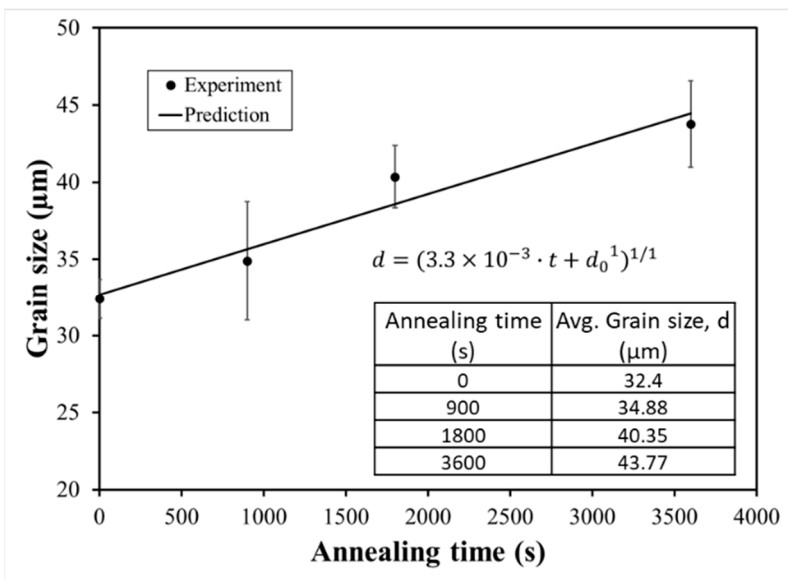

**Figure 2.** Grain growth of 253 MA ASS during annealing at 1100 °C with variations in the annealing time.

Figure 3a–e show the microstructures of 253 MA ASS after cold rolling followed by annealing at 1100 °C with variations in the annealing time. Precipitation was visible on the microstructure of 253 MA ASS after cold rolling and annealing. These precipitates were distributed in the austenite matrix and grain boundaries, which can cause a pinning effect so that grain growth increases slightly with annealing time [27]. The 253 MA ASS after annealing at 1800 s showed more precipitation than other annealing times. In addition, high-reduction cold rolling up to 53% in 253 MA ASS resulted in many strains inducing the formation of martensite. During annealing, the transformation of martensite to austenite reversion occurs slowly. It contributes to inhibiting the rapid growth of austenite grains [28].

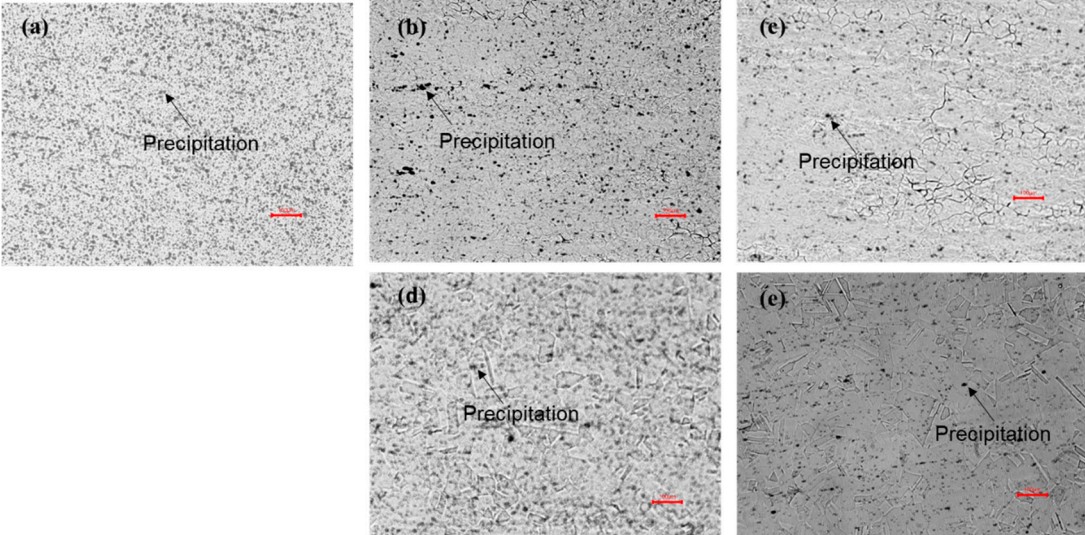

**Figure 3.** Microstructures of 253 MA ASS after (**a**) cold rolling and then (**b**) annealing at 1100 °C with annealing times of (**c**) 900; (**d**) 1800; (**e**) 3600 s with 100× magnification.

Figure 4a–e show observations of precipitates using SEM. The precipitates appeared spherical with relatively small and large sizes in the austenite matrix and grain boundaries.

The diameter size of the precipitate changed after cold rolling and annealing over 900, 1800, and 3600 s, respectively, as follows: 9.8, 8.8, 13.2, and 10.4 μm. The number of precipitates per area increased with the annealing time until 1800 s, after which the number of precipitates decreased, as shown in Figure 5. The presence of precipitates at the austenite grain boundaries causes a pinning effect, resulting in sluggish grain growth [27].

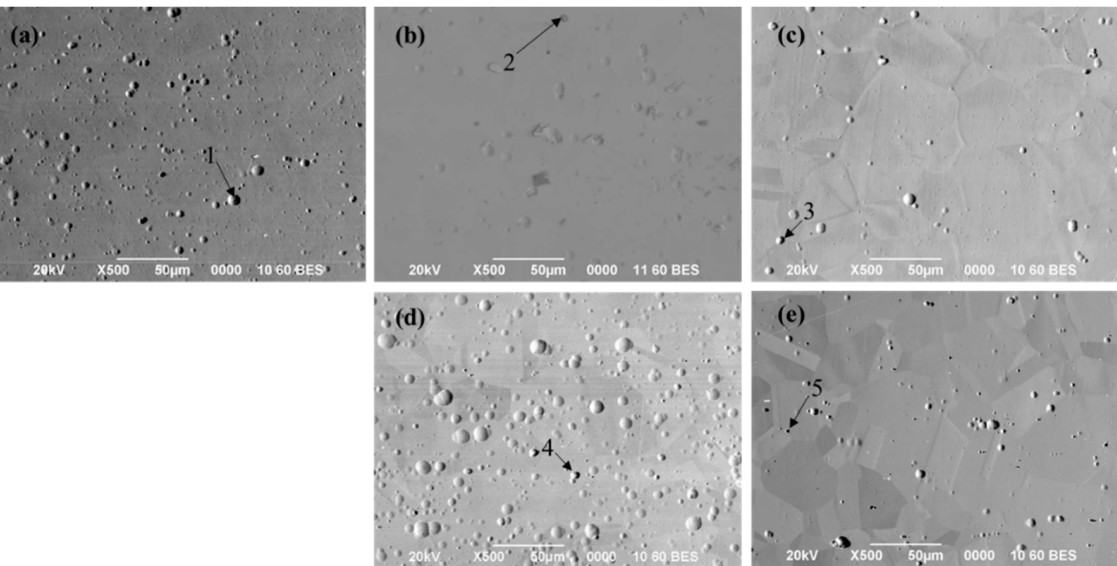

**Figure 4.** Observation of precipitates (indicated by arrows) on 253 MA ASS from (**a**) cold rolling-arrow no. 1, (**b**) annealing at 1100 °C-arrow no. 2 and with annealing times of (**c**) 900 s-arrow no. 3; (**d**) 1800 s-arrow no. 4; (**e**) 3600 s-arrow no. 5 using SEM and EDAX analyses.

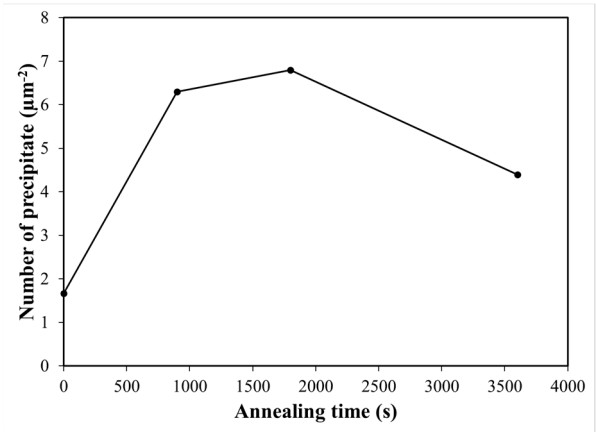

**Figure 5.** Effect of annealing time on the number of precipitates per area.

Table 2 shows a point-by-point EDAX analysis of the precipitates, shown by the arrows for each annealing time in Figure 4. In this table, the elements C, Cr, Ni, Ce, and Fe may form metal carbide ($M_{23}C_6$) precipitates in the austenite grains. With increasing annealing time, the percentage of carbon atoms decreased slightly. This indicates that the longer annealing time caused metal carbide precipitates to dissolve into the austenite grains. However, in 253 MA ASS with an annealing time of 900 s, the carbon atoms in the precipitate could not be detected. This was probably due to the relatively little dissolution of carbon atoms in the austenite matrix. In addition, the presence of chrome and nitrogen elements in 253 MA ASS is likely to form $Cr_2N$ precipitates in the microstructure, which contribute to inhibiting grain growth [29].

**Table 2.** EDAX analysis of the precipitates.

| No. | Sample | % Atom | | | | | | | |
|---|---|---|---|---|---|---|---|---|---|
| | | **C** | **O** | **Si** | **Cr** | **Mn** | **Fe** | **Ni** | **Ce** |
| 1 | Cold Rolled | 33.26 | - | 1.21 | 15.04 | - | 43.76 | 6.73 | - |
| 2 | 1100 °C | 30.7 | 4.42 | | 14.71 | 2.09 | 43.1 | 4.97 | - |
| 3 | 1100 °C—900 s | - | - | - | 24.35 | 1.45 | 65.08 | 9.11 | - |
| 4 | 1100 °C—1800 s | 28.48 | 8.08 | - | 16.02 | - | 42 | 5.45 | - |
| 5 | 1100 °C—3600 s | 27.29 | 6.21 | - | 16.1 | - | 43.21 | 6.27 | 0.45 |

*3.2. Effect of Grain Size on the Mechanical Properties*

Figure 6 shows the tensile test results curve of 253 MA ASS after cold rolling and annealing at 1100 °C. In this figure, cold-rolled steel had a higher yield strength and ultimate tensile strength than annealed steel, while for the 253 MA ASS, after annealing at 1100 °C each time, the yield strength and ultimate tensile strength values were not significantly different. Annealed steel was lower in strength than cold-rolled steel was, but it had a higher engineering strain value than cold-rolled steel did. In addition, the increasing annealing time reflected the coarser grain size increasing the engineering strain or ductility values. However, 253 MA ASS annealed for 900 s showed higher engineering strain values than those annealed for 1800 s. The dissolution % atom of Cr, Mn, Ni, and Fe atoms in 253 MA ASS with an annealing time of 900 s was higher than those for the other annealing times (Table 2), which appeared in the austenite grains as intermetallic precipitates, resulting in reducing the engineering strength slightly and increasing the engineering strain slightly.

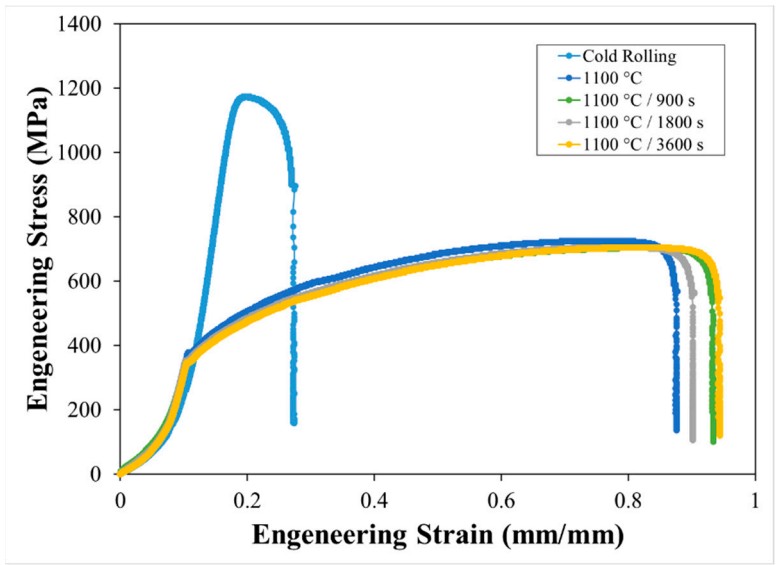

**Figure 6.** Engineering stress vs. engineering strain of 253 MA ASS after cold rolling and annealing with variations in the annealing time.

Figure 7 shows the effect of annealing time on the value of the micro-Vickers hardness. In this figure, the hardness of 253 MA ASS decreased drastically after the steel was cold-rolled and decreased slightly after the steel was annealed. The longer the annealing time, the coarser the grain, which results in softening of the steel. However, this figure shows that the hardness of 253 MA ASS with an annealing time of 3600 s was slightly higher than that for the annealing time of 1800 s. This was probably due to the number of precipitates at the annealing time of 3600 s (in Figure 5) being lower than for the annealing times of 900 and 1800 s. For comparison, at the same cold-rolling percentage and annealing temperature and a longer annealing time, the hardness of cold-rolled steel in this study had a hardness

value greater than the hardness value of American Iron and Steel Institute (AISI) 904L super austenitic stainless steel in the study by Stornelli et al. [30]. This was probably due to the high nitrogen content in its chemical composition and the presence of micro-alloying cerium deposited in the austenite matrix, which can increase the hardness of 253 MA ASS.

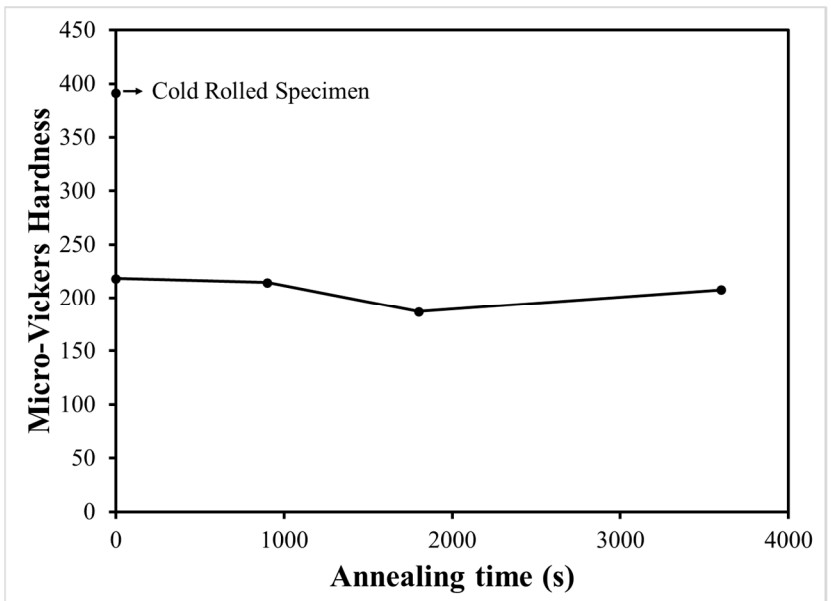

**Figure 7.** Effect of annealing time on the value of the micro-Vickers hardness.

Figure 8 shows the relationship between yield strength and grain size of $d^{-\frac{1}{2}}$ 253 MA ASS using the Hall–Petch equation. The Hall–Petch equation is shown in Equation (2):

$$\sigma_y = \sigma_0 + k' \cdot d^{-1/2} \qquad (2)$$

where $\sigma_y$ is the yield strength, $\sigma_0$ and $k'$ are experimental constants, and $d$ is the grain size. The higher the value of $d^{-\frac{1}{2}}$, the finer the grain sizes obtained. The experimental results were almost linear with the prediction results using Equation (2). The fitted line showed a slope of 28.5 MPa μm$^2$ and a flow stress of 223.2 MPa at an infinite grain size. In this figure, the yield strength value increased slightly with the degree of grain fineness. When compared with previous studies using 316L steel, it has a yield strength value of approximately 290–301 MPa at room temperature at a grain size $d^{-\frac{1}{2}}$ of approximately 0.08–0.16 μm$^{-\frac{1}{2}}$ [31]. In this study, 253 MA ASS had a yield strength value of approximately 350–400 MPa at room temperature at a grain size $d^{-\frac{1}{2}}$ of approximately 4.4–5.6 mm$^{-\frac{1}{2}}$ or 0.14–0.17 μm$^{-\frac{1}{2}}$. This confirms a grain boundary strengthening effect that was influenced by differences in chemical composition, especially the nitrogen content and the micro-alloying cerium, and this result agrees with previous studies [32,33].

Figure 9 shows the effect of grain size on the elongation of 253 MA ASS. This figure shows that the elongation value fluctuates with increasing degrees of fineness of grain size, $d^{-\frac{1}{2}}$. Although fluctuating, the difference in elongation percentage was not significantly different with increasing $d^{-\frac{1}{2}}$ grain fineness. Therefore, the presence of precipitates in austenite grains caused an insignificant difference in elongation, compared with 18Mn18Cr0.6N steel in previous studies, which had an elongation value of 43% [34]. The elongation value of 253 MA ASS was higher than that of 18Mn18Cr0.6N due to the differences in the chemical composition, percentage of cold rolling, and annealing temperature.

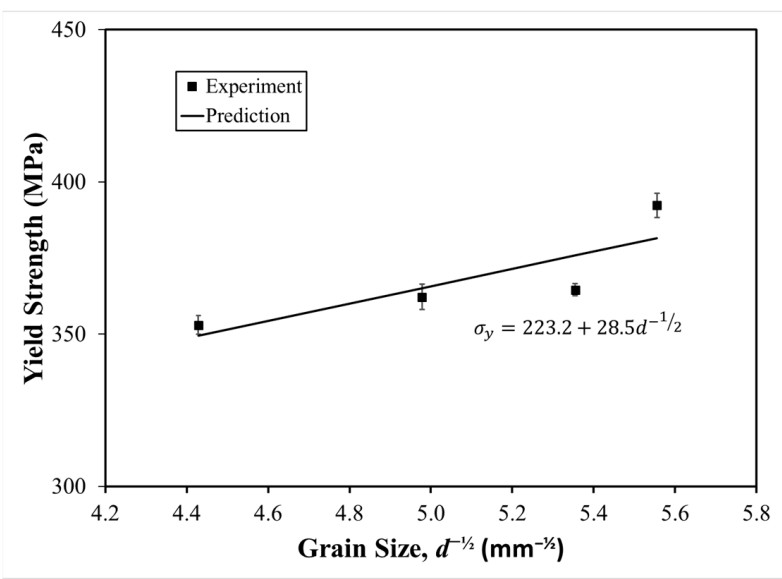

**Figure 8.** The relationship between yield strength and grain size and its prediction using the Hall–Petch equation.

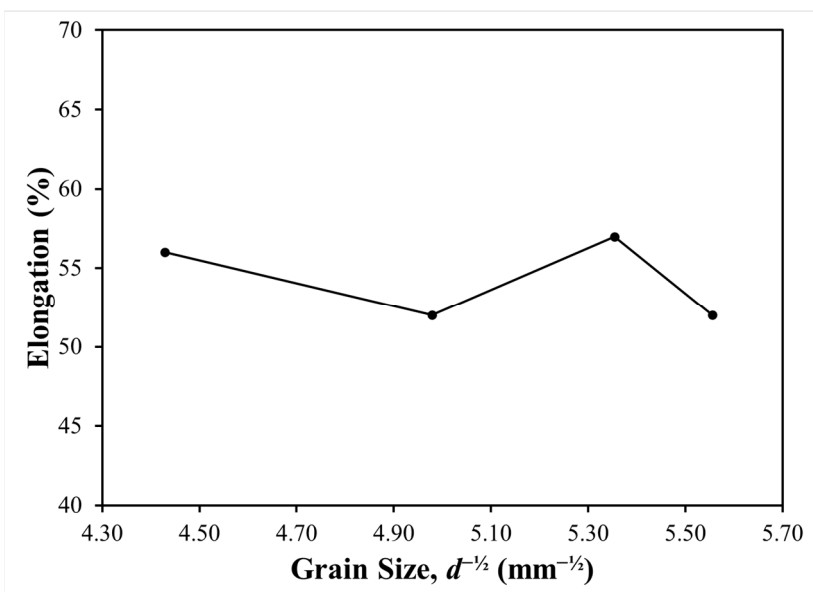

**Figure 9.** Effect of grain size on elongation.

Figure 10 shows the effect of grain size on the strain-hardening exponent. The calculation of the strain-hardening exponent and constant refers to ASTM E646—method B, as shown in Equation (3):

$$\sigma = K\varepsilon^n \tag{3}$$

where $\sigma$ is the true stress (ksi), $K$ is a constant, $\varepsilon$ is the plastic component of true strain, and n is the strain-hardening exponent. In this figure, the strain-hardening exponent, $n$, decreased with the grain size, $d^{-\frac{1}{2}}$, as did the value of the constant, K. This was because the more refined the grain size, the higher the yield strength, but the strain decreased so that the value of the strain-hardening exponent, $n$, decreased. Xu et al. [35], in their research on the effect of grain refinement on strain-hardening, found that the finer the austenite grain size, the lower the strain-hardening rate. This result agrees with the previous study.

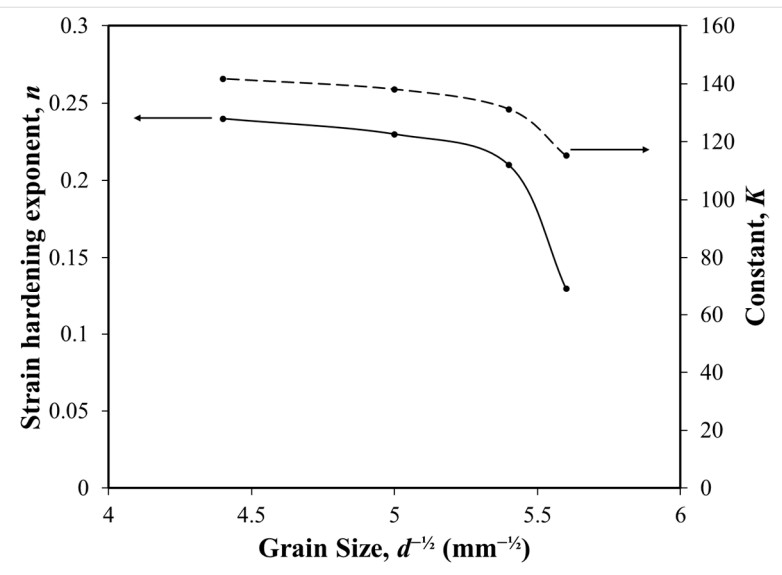

**Figure 10.** Effect of grain size on the strain-hardening exponent and constant K.

Figure 11 shows the effect of grain size on the micro-Vickers hardness. In this picture, the hardness value increased slightly with the fineness of the grain size, $d^{-1/2}$, even though at a grain size $d^{-\frac{1}{2}}$ of 5 mm$^{-\frac{1}{2}}$, the hardness value decreased. This was probably due to the dissolution of the elements Cr, Mn, Fe, and Ni as precipitates in austenite grains cause the hardness value of 253 MA ASS at a grain size $d^{-\frac{1}{2}}$ of 5 mm$^{-\frac{1}{2}}$ to decrease.

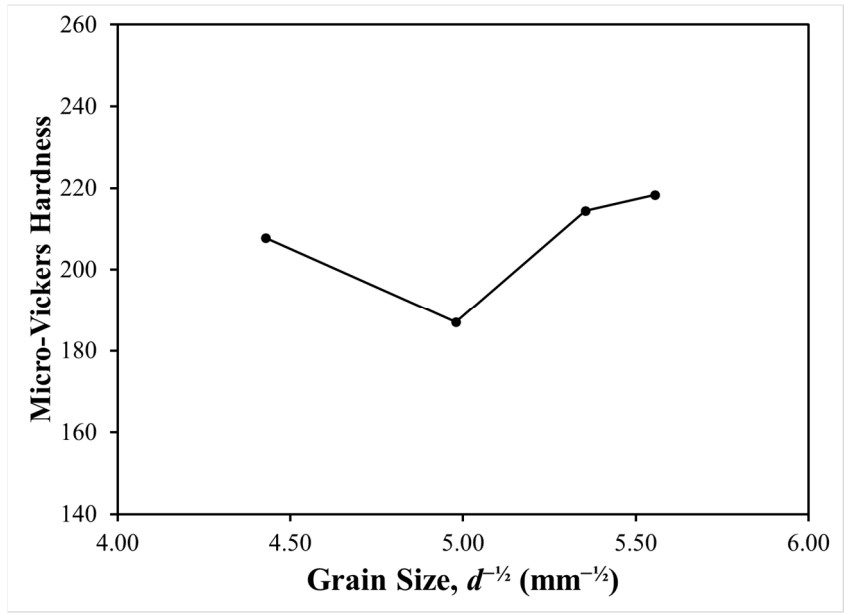

**Figure 11.** Effect of grain size on the micro-Vickers hardness.

Figure 12a–e show SEM observations on tensile test fractures at 253 MA ASS after cold rolling and annealing at 1100 °C with variations in the annealing time. In these images, the fracture that occurred in the steel after cold rolling and annealing was a ductile fracture, which is characterized by a dimple shape on the fracture surface. With an increasing annealing time, a coarser grain size occurred and the maximum dimple size increased, as shown in Figure 13.

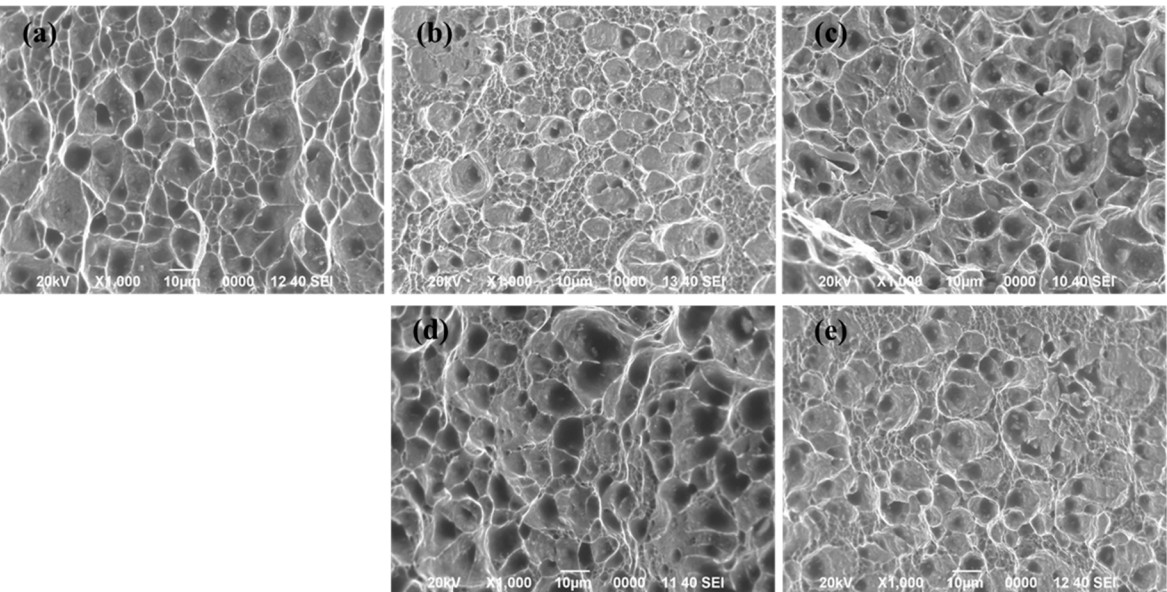

**Figure 12.** SEM observations of the tensile test fractures of 253 MA ASS after (**a**) cold rolling and (**b**) annealing at 1100 °C with annealing times of (**c**) 900; (**d**) 1800; (**e**) 3600 s.

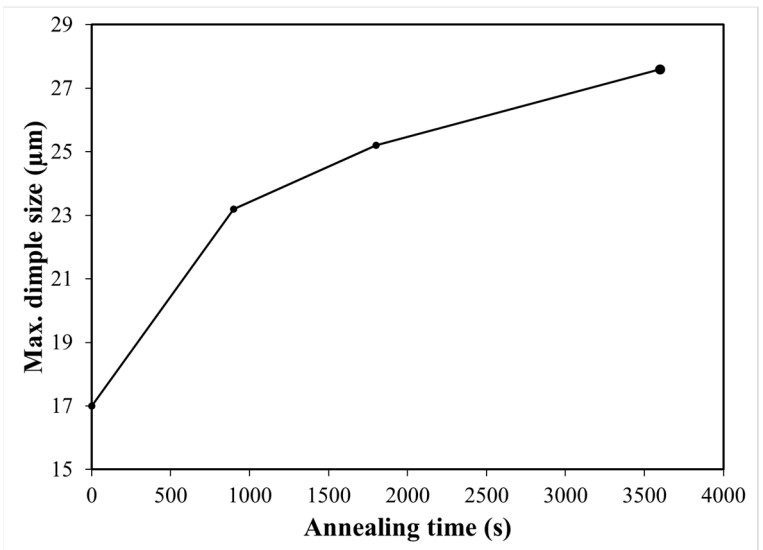

**Figure 13.** Effect of annealing time on the maximum dimple size on the tensile fracture surface.

### 3.3. Effect of Grain Size on Creep Rupture Properties

Figure 14 shows the effect of annealing time on the creep rupture behavior of 253 MA ASS at a temperature of 700 °C and a load of 150 MPa. Primary, secondary, and tertiary creep stages occurred for all of the annealing times. The primary creep stage at the annealing time of 3600 s showed a higher elastic strain than the other annealing times. The secondary creep stage at the annealing time of 1800 s was longer than for other the annealing times, which was caused by a high resistance to creep rupture.

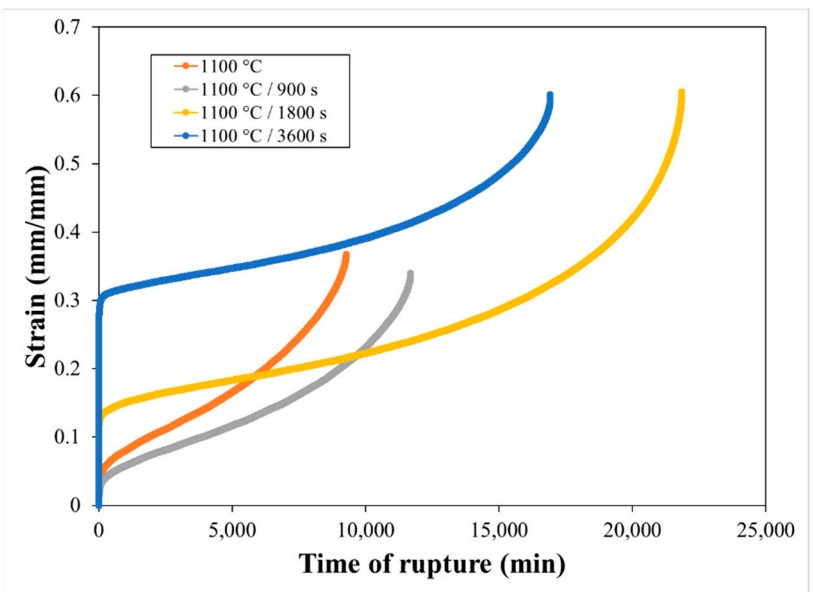

**Figure 14.** Effect of annealing time on the creep rupture behavior of 253 MA ASS at 700 °C and a load of 150 MPa.

Table 3 shows the creep parameter of 253 MA after annealing at 1100 °C with variations in annealing time. This table shows that the highest elastic creep strain of 0.306 mm/mm was found in 253 MA ASS after annealing for 3600 s, while for the annealing time of 1800 s, the highest fracture strain and time of fracture occurred. The lowest creep rate of $7.95 \times 10^{-6}$ s$^{-1}$ was found at 253 MA ASS with an annealing time of 1800 s.

**Table 3.** Creep parameter of 253 MA ASS after annealing at 1100 °C with variations in the annealing time.

| Creep Parameter | Annealing Time (s) | | | |
|---|---|---|---|---|
| | 0 | 900 | 1800 | 3600 |
| $\varepsilon_{elastic}$ (mm/mm) | 0.055 | 0.039 | 0.135 | 0.306 |
| $\varepsilon_{fracture}$ (mm/mm) | 0.367 | 0.339 | 0.605 | 0.601 |
| $t_{fracture}$ (min) | 9270.4 | 11,683.6 | 21,846.1 | 16,924.4 |
| Creep rate (s$^{-1}$) | $22.1 \times 10^{-6}$ | $16.8 \times 10^{-6}$ | $7.95 \times 10^{-6}$ | $8.29 \times 10^{-6}$ |

Figure 15 shows a comparison of grain sizes before and after the creep rupture test. In this figure, the grain size increased slightly after the creep rupture test. This indicates that the grain growth of 253 MA ASS was still ongoing during the creep rupture test at a temperature of 700 °C and a load of 150 MPa. The grain size before and after the creep rupture test showed almost the same value at 253 MA ASS after being annealed for 3600 s.

Figure 16a–c show the relationship between rupture time and initial grain size, micro-Vickers hardness, and yield strength. Figure 14a shows that the time of rupture increased with grain size, from approximately 32 to approximately 40 µm, while at grain sizes greater than approximately 40 µm, the time of rupture decreased. Figure 14b shows that the higher the micro-Vickers hardness value, the lower the rupture time. Figure 14c shows that the time of rupture increased as the yield strength increased from approximately 350 to approximately 360 MPa, while the yield strength over approximately 360 MPa decreased the rupture time. Based on the results of the linear regression, it was found that there was a strong relationship between time of rupture with initial grain size and micro-Vickers hardness with an $R_2$ value of approximately 0.9.

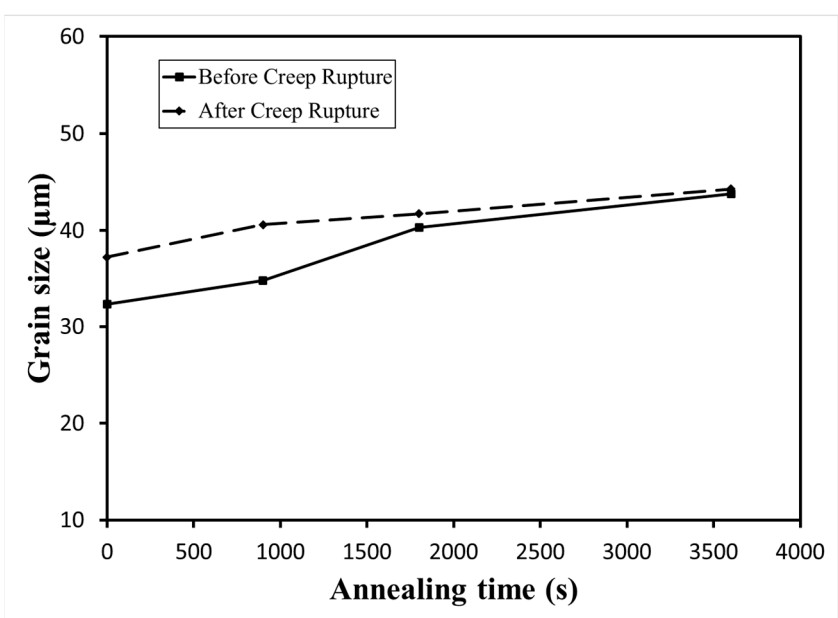

**Figure 15.** Comparison of grain sizes before and after the creep rupture test.

Figure 17a–d show the microstructure of 253 MA with variations in the annealing time of the gauge length after the creep rupture test at a temperature of 700 °C and a load of 150 MPa. The grain size of 253 MA ASS at each annealing time was coarser after the creep rupture test, which consisted of large and small grains. 253 MA ASS with an annealing time of 1800 s (Figure 17c) after the creep rupture test showed austenite grains consisting of larger and smaller grain sizes. This indicates that abnormal or discontinuous growth of austenite grains occurred during the creep rupture test. This grain growth was characterized by rapidly growing grains in a matrix of small grains, which results in a bimodal grain size distribution, causing the creep fracture time to be longer than 253 MA with other annealing times. The results of this study are different from the results of previous studies that stated that the larger the grain size, the higher the creep rupture time [13,36]. Annealing twins were seen in the austenite grains at each annealing time, where the size followed the austenite grain size. After the creep rupture test, the gauge length section did not show any creep in the austenite grains. At each annealing time, concentrated precipitates at the austenite grain boundaries could initiate intergranular cracks at 253 MA ASS.

Figure 18a–d show the microstructure of 253 MA with variations in annealing time near the fracture after the creep rupture test with a 500× magnification. After the creep rupture test, the near section of the fracture did not show any elongation in the austenite grains. When compared with research conducted by Liu et al. [15] and Manokaran et al. [37], Ni–Mo–Cr–Fe superalloy (Hastelloy N) and austenitic stainless-steel SS321 showed elongated grains near the fracture after creep rupture tests were performed. The precipitation is relatively concentrated at the grain boundaries of the triple junction so that it can trigger intergranular cracking when the load is applied. In addition, at 253 MA ASS at annealing times of 1800 and 3600 s after the creep rupture test, there was crack propagation from the grain boundaries to the grain matrix due to the interconnection between the precipitates at the grain boundaries and the austenite matrix, which triggers transgranular cracking when the load is applied. Transgranular failure was also found by Monteiro et al. [38] in their research on the creep rupture mechanism of AISI 316 austenitic stainless steel and by Sinya et al. [39] in their study of the mechanism of creep fracture in AISI 316 after undergoing creep rupture for 100,000 h.

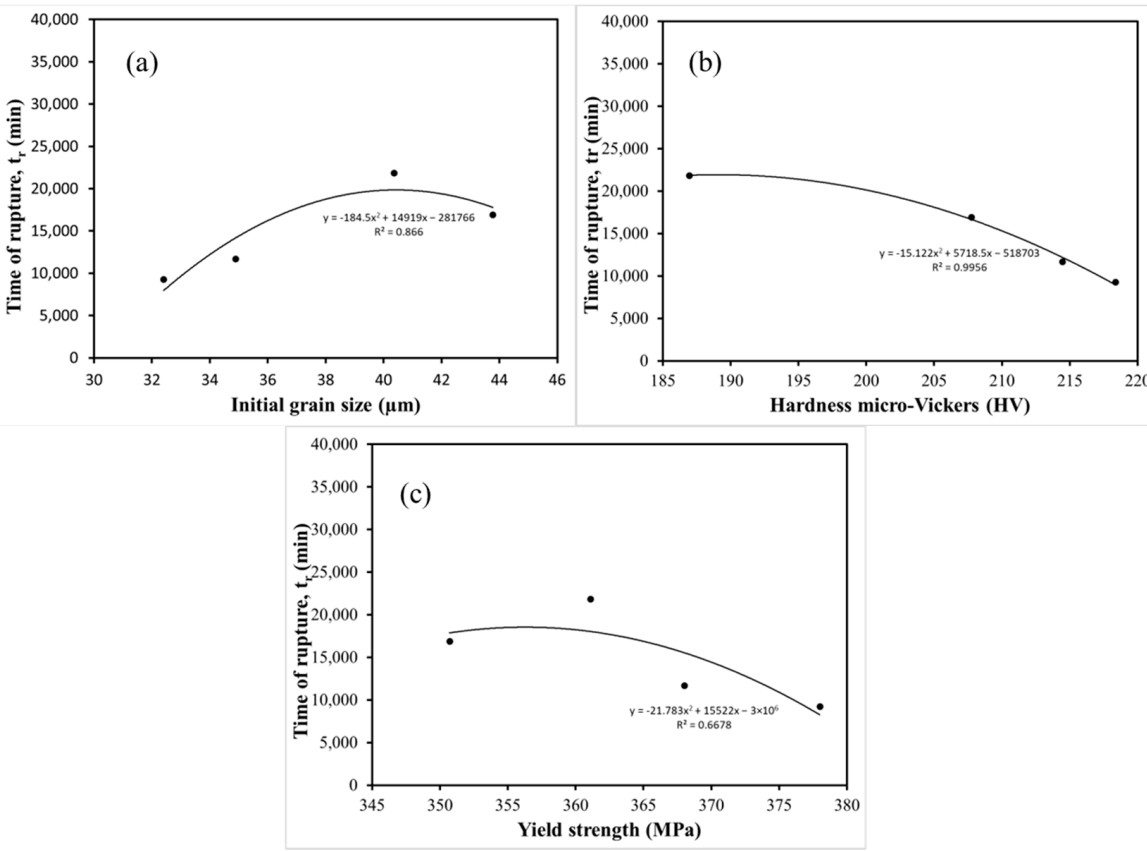

**Figure 16.** The relationship of time of rupture to (**a**) initial grain size; (**b**) micro-Vickers hardness; (**c**) yield strength.

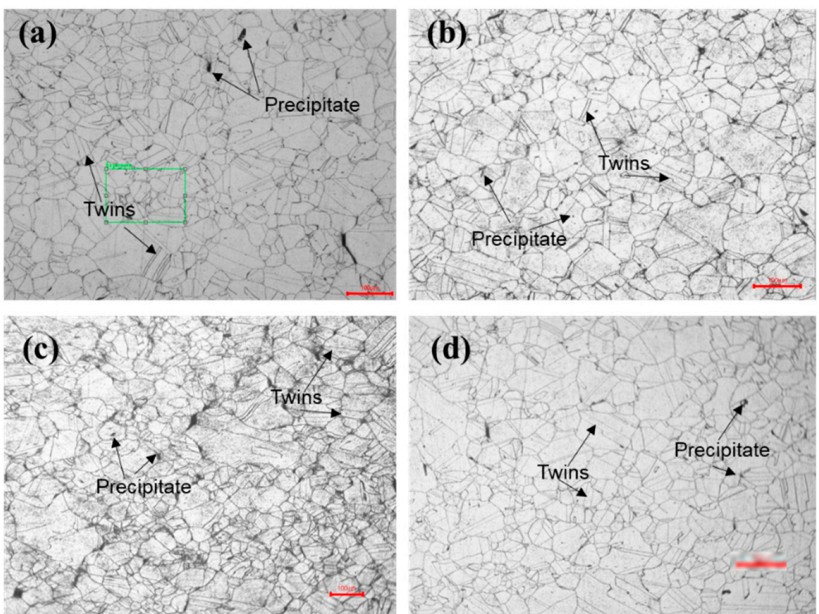

**Figure 17.** Microstructures of 253 MA ASS at annealing times of (**a**) 0; (**b**) 900; (**c**) 1800; (**d**) 3600 s at gauge length after creep rupture test with a 100× magnification.

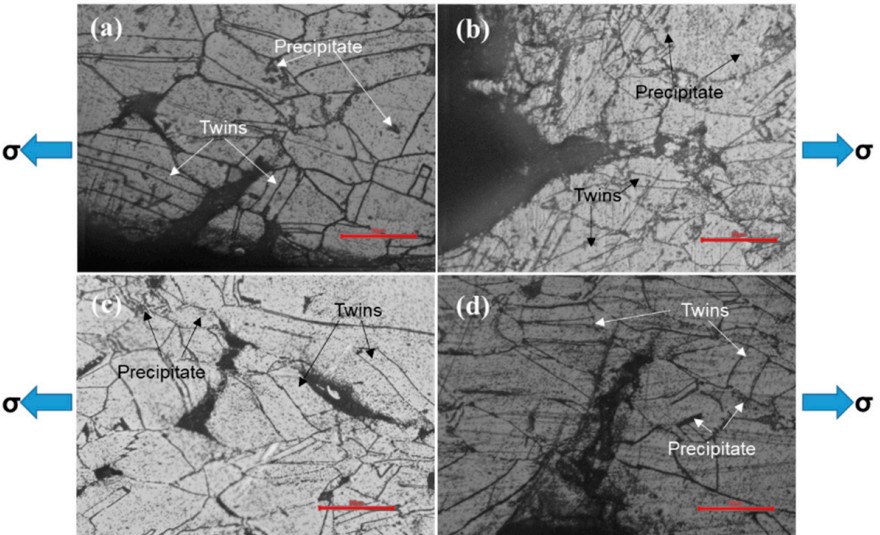

**Figure 18.** Microstructural observations of the near section of fracture at 253 MA without (**a**) annealing time and annealing times of (**b**) 900; (**c**) 1800; (**d**) 3600 s after creep rupture test with a $500\times$ magnification.

Figure 19a–d show the SEM observations on the near section of fracture of 253 MA ASS with variations in the annealing time after creep rupture tests with a 500X magnification. This figure shows that 253 MA ASS without annealing and with an annealing time of 900 s experienced intergranular fracture after the creep rupture test. Meanwhile, 253 MA ASS with annealing times of 1800 and 3600 s experienced mixed-mode intergranular and transgranular fractures. This fracture mode was triggered by micro-cavity coalescence and intergranular cracking in the grain boundary under creep rupture test [40]. Figure 20 shows that the average creep cavity size of 253 MA ASS with an annealing time of 1800 s was less than that of 253 MA ASS with an annealing time of 3600 s; therefore, the creep fracture time of 253 MA ASS with an annealing time of 1800 s was longer than for 253 MA ASS with an annealing time of 3600 s.

Figure 21 shows a comparison of the micro-Vickers hardness values before and after the creep rupture test on the variation in austenite grain size, $d^{-\frac{1}{2}}$. The measurement of the micro-Vickers hardness after the creep rupture test was carried out on the gauge length section of 253 MA ASS. The hardness value after the creep rupture test was higher than before the creep rupture test when the austenite grain size, $d^{-\frac{1}{2}}$, reached above approximately 5 mm$^{-\frac{1}{2}}$. Although the grain size of the sample after creep rupture was slightly coarser than the grain size of the sample before creep rupture (Figure 13), the hardness value of the sample after creep rupture was slightly higher than that of the sample before creep rupture at austenite grain fineness levels above 5 mm$^{-\frac{1}{2}}$. The difference in the hardness values between steels before and after creep rupture may be due to grain boundary hardening caused by changes in grain size during the creep rupture test [41].

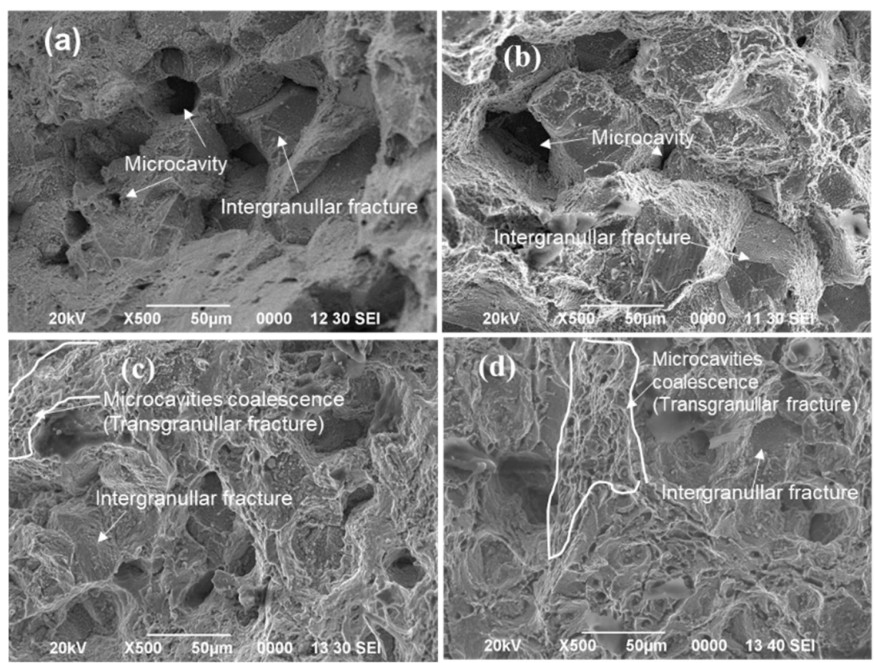

**Figure 19.** SEM observations of the fracture surface of the 253 MA ASS (**a**) without annealing and with annealing times of (**b**) 900; (**c**) 1800; (**d**) 3600 s after creep rupture test with a 500× magnification.

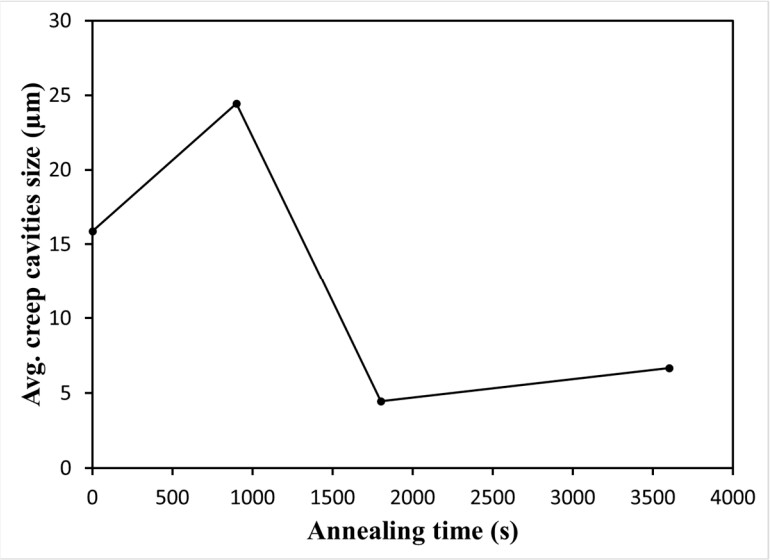

**Figure 20.** Effect of annealing time on the average creep cavity size (μm).

Figure 22 shows the grain size's, $d^{-\frac{1}{2}}$, effect on creep ductility. The lowest creep ductility was found at 253 MA ASS with a grain size $d^{-\frac{1}{2}}$ of approximately 5 mm$^{-\frac{1}{2}}$. At this grain size, $d^{-\frac{1}{2}}$, the creep rupture time was the highest compared to those of the other grain sizes, $d^{-\frac{1}{2}}$, and had a low creep ductility. The low creep ductility caused the elongation of the steel to decrease, which was indicated by the presence of creep cavities at the fracture [42].

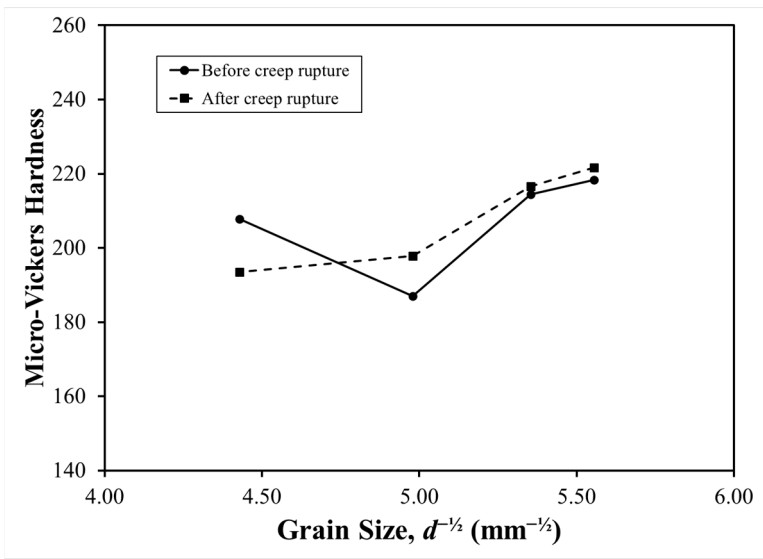

**Figure 21.** Comparison of hardness vs. grain size before and after the creep rupture test.

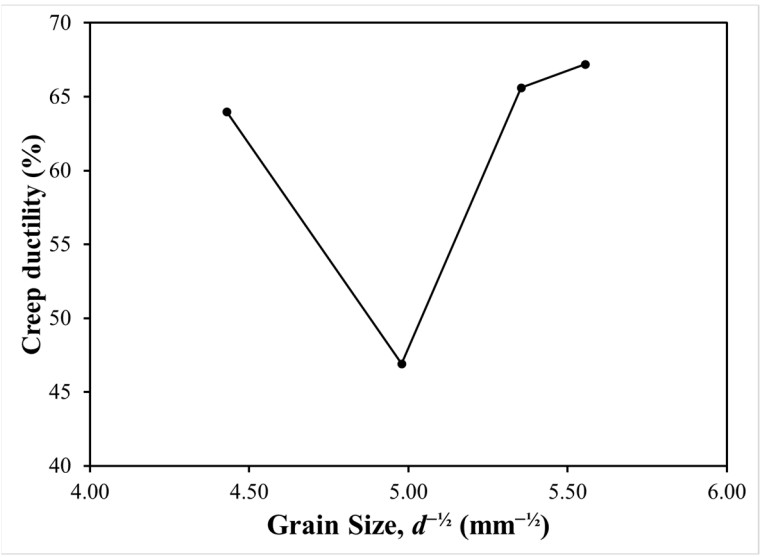

**Figure 22.** Effect of grain size on creep ductility.

The low value of the micro-Vickers hardness in steel with a grain size ($d^{-\frac{1}{2}}$) of approximately 5 mm$^{-\frac{1}{2}}$ or with an annealing time of 1800 s was probably due to the large number of precipitates per area formed on the austenite matrix. Then, the precipitate can dissolve and reappear in the austenite matrix and the grain boundaries after the steel is subjected to a creep rupture test at a temperature of 700 °C and a constant load of 150 MPa. At this constant temperature and load, the precipitate can act as a trigger for the nucleated micro-cavities. If the precipitate is present in the austenite matrix, it will cause the micro-cavity to nucleate and propagate until transgranular fracture occurs. Meanwhile, intergranular fractures can be caused by the presence of precipitates at the austenite grain boundaries so that micro-cavities can form and propagate at the grain boundaries [43]. Figure 19c–d show the 253 MA ASS fractures in the mixed transgranular and intergranular modes after the creep rupture test. The mixed-mode fracture was indicated by the presence of micro-cavities coalescence and intergranular cracking on the fracture surface. Mixed-mode transgranular and intergranular fractures of 253 MA ASS with a grain size ($d^{-\frac{1}{2}}$) of approximately 5 mm$^{-\frac{1}{2}}$ showed relatively little coalescence of micro-cavities than 253 MA

ASS with a grain size ($d^{-\frac{1}{2}}$) of approximately 4.4 mm$^{-\frac{1}{2}}$. A possible cause is that the 253 MA ASS with a grain size ($d^{-\frac{1}{2}}$) of approximately 5 mm$^{-\frac{1}{2}}$ had a high creep rupture time.

## 4. Conclusions

In this study, the effect of grain size on the mechanical properties and creep rupture properties of 253 MA ASS was studied, and the following can be concluded:

(1) Grain growth at a temperature of 1100 °C showed a slight increase with annealing time. A fine grain size can increase yield strength and hardness, and the ultimate strength was almost constant, with fluctuating elongation and decreased strain hardening;

(2) $M_{23}C_6$ and intermetallic precipitates and micro-alloying in austenite grains resulted in slow grain growth that affected the mechanical properties at room temperature. Ductile fracture occurred in the cold rolling and annealing samples. The grain size influenced the size of the dimple at the fracture surface. The larger the grain size, the larger the resulting dimple size;

(3) Creep rupture at a temperature of 700 °C and a load of 150 MPa showed that the austenite grain growth continued even though the difference between the grain size before and after the creep rupture test was not significant;

(4) A higher creep rupture time and a lower creep ductility were found on the grain size of approximately 40 μm. However, this grain size had a low value of hardness and yield strength;

(5) The normal grain size distribution during the creep rupture test resulted in a fast rupture time with intergranular fractures occurring in steels with an initial grain size below 40 μm. In comparison, the bimodal grain size distribution during the creep rupture test increased the rupture time, resulting in mixed-mode intergranular and transgranular fractures in steel with an initial grain size above 40 μm;

(6) The difference in grain size before and after the creep rupture test caused grain boundary hardening, resulting in a higher steel hardness value after the creep rupture test.

**Author Contributions:** Conceptualization, M.S.A. and E.S.S.; methodology, M.S.A. and E.S.S.; validation, M.S.A., R.R.W., L.B.A.P., A.A.A., E.M. and E.S.S.; formal analysis, M.S.A.; investigation, M.S.A.; resources, M.S.A.; data curation, M.S.A., L.B.A.P. and A.A.A.; writing—original draft preparation, M.S.A.; writing—review and editing, M.S.A. and E.S.S.; visualization, M.S.A., E.M. and E.S.S.; supervision, M.S.A., E.M. and E.S.S.; project administration, M.S.A. and E.S.S.; funding acquisition, M.S.A. and E.S.S. All authors have read and agreed to the published version of the manuscript.

**Funding:** This research was funded by the PUTI Doktor 2020 (grant number: NKB-3355/UN2.RST/HKP.05.00/2020) and BRIN Research (grant number: SK 197/H/2019).

**Data Availability Statement:** Not applicable.

**Acknowledgments:** The authors would like to thank P.T. Cahaya Bina Baja-Sandvik, who supported the procurement of the 253 MA austenitic stainless steel.

**Conflicts of Interest:** The authors declare no conflict of interest.

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
