# Peer review of "Effect of Grain Size on Mechanical and Creep Rupture Properties of 253 MA Austenitic Stainless Steel"

_metals, doi:10.3390/met12050820_

Round 1
Reviewer 1 Report
t is a traditional idea devoted to various grain sizes on mechanical, namely ‘Effect of Grain Size on Mechanical and Creep Rupture Properties of 253 MA Austenitic Stainless Steel’. The paper could be published after revisions as follow:
- The factors that affect the grain growth are temperature first and then time. This article investigates the effect of grain size on mechanical properties, why it doesn’t choose the most effective method of grain control.
- Grain changes are not well observed for short time intervals during heat treatment. The grain size statistics in Fig.3 are not convincing enough, which is the most critical reason for almost no difference in mechanical properties.Subsequent analyses based on this uncertain statistical result are futile.
- Why heat treatment in H2 atmosphere, which is wasteful and dangerous.
Reviewer 2 Report
Reviewer’s comments: “Effects of grain size on mechanical and creep rupture properties of 253 MA austenite stainless steel” by Moch. Syaiful Anwar et. al.
In this article, effects of grain size on mechanical and creep rupture properties of 253 MA austenite stainless steel are researched. However, the article needs the major reversion to be published because the analysis and the processing of the experimental data are due to the lack of qualification.
Author had better improve as follows:
(1) The following figures should be completely revised.
In Fig2, the error bar needs to seek accuracy.
In Fig 3, the shapes of precipitations in the Fig3 (a-e) are not clear. Authors had better show the clear shapes of the precipitation.
In Fig4, the correspondence of photographs in the pictures and the explanation of the text are not clear. In addition, the relations between numbers of precipitations and annealing time should be quantified. What is the arrow in the figure?
In Fig6, what is discrepancy between the hardness at annealing time 0 and that of the cold rolled specimen?
In Fig7, this figure is quite important for Hall-Petch relation. Therefore, author should be rewritten using the expanding scale of Yield stress and errors bar.
In Fig11 and 17, the dimple, cup, cone shape and micro-void are not clear in the photograph. Author should clearly show those in the photos, for example, drawing the schematic figure.
In Fig15 and 16, in order to clarify the explanation of the text this picture needs the schematic figure which are included by the observed precipitations and twins.
(2) Necessary of quantifications of data
In order to gain a quantitative grasp of the data, the numbers of dimple, cup, cone shape, micro-void, precipitation and micro-crack should be demonstrated as a function of annealing time.
(3) On the discussion
Author should minutely and quantitatively discuss the results of Fig 18 and Fig 19, because these data are more important in this article. Especially, author should quantitatively explain the reason why both of the hardness and the creep ductility take the minimum at 5 of the grain size.
(4) On English It is poor written in English and there is a room for improvement.
Reviewer 3 Report
This manuscript is presents the analysis of Grain size effect to mechanical and creep rupture properties for stainless steel.
The methodology and obtained result are clear, but I have some questions as follows;
(1) How did you measure grain size (32.4, 34.88, 40.35, 43.77 um) of the 253 MA ? Are there any photos indicating the grain size of 253 MA ? (like as Fig.3 or 4)
(2) Why does the composition of the atoms except for carbon in 253MA depend on the annealing time ? (as shown in Table 2)
(3) As described by the authors (L178-181), only the component for 900s annealing is deviated. Is this result repeatable ?
(4). I am confusing the x-axis in Fig. 7-10, 18, 19. Why d^-1/2 are around ~4.5~5.5, for d= 32.4~43.77 ?
Reviewer 4 Report
That's OK for the publication just a few sentences are required to explain, where the real industrial application of this research is. The author can put at the end of the introduction. A few typing errors in the text and I think, it is necessary a little bit simplify the conclusions
Round 2
Reviewer 1 Report
Accept
Author Response
Dear Reviewer, I have done correction of English language and style using MDPI English service editing. Please find the certificate in the attachment

Reviewer 2 Report
There are many grammatical errors and miss-typesetting; for instance, Line no. 90, 91, 173, 473, 475, 476, 480 and so on. Author should improve them.
Author Response
Dear Reviewer, I have done correction of English language and style using MDPI English service editing. Please find the certificate in the attachment file.
